# Overcoming myelosuppression due to synthetic lethal toxicity for FLT3-targeted acute myeloid leukemia therapy

Alexander A Warkentin[1], Michael S Lopez[1], Elisabeth A Lasater[2], Kimberly Lin[2], Bai-Liang He[3], Anskar YH Leung[3], Catherine C Smith[2]*, Neil P Shah[2]*, Kevan M Shokat[1]*

[1]Department of Cellular and Molecular Pharmacology, Howard Hughes Medical Institute, University of California, San Francisco, San Francisco, United States; [2]Division of Hematology and Oncology, University of California, San Francisco, San Francisco, United States; [3]Division of Haematology, Department of Medicine, Li Ka Shing Faculty of Medicine, University of Hong Kong, Pokfulam, Hong Kong

**Abstract** Activating mutations in FLT3 confer poor prognosis for individuals with acute myeloid leukemia (AML). Clinically active investigational FLT3 inhibitors can achieve complete remissions but their utility has been hampered by acquired resistance and myelosuppression attributed to a 'synthetic lethal toxicity' arising from simultaneous inhibition of FLT3 and KIT. We report a novel chemical strategy for selective FLT3 inhibition while avoiding KIT inhibition with the staurosporine analog, Star 27. Star 27 maintains potency against FLT3 in proliferation assays of FLT3-transformed cells compared with KIT-transformed cells, shows no toxicity towards normal human hematopoiesis at concentrations that inhibit primary FLT3-mutant AML blast growth, and is active against mutations that confer resistance to clinical inhibitors. As a more complete understanding of kinase networks emerges, it may be possible to define anti-targets such as KIT in the case of AML to allow improved kinase inhibitor design of clinical agents with enhanced efficacy and reduced toxicity.

*For correspondence: csmith@medicine.ucsf.edu (CCS); nshah@medicine.ucsf.edu (NPS); Kevan.Shokat@ucsf.edu (KMS)

## Introduction

Kinase inhibitors are among the fastest growing new class of therapeutics for treating cancer, with 25 new kinase inhibitors approved by the FDA in the last 14 years (*Mullard, 2014*). Since these agents almost exclusively target a kinase's highly conserved ATP binding pocket, achieving selectivity is problematic. A limiting feature of kinase inhibitors is that their ability to target multiple wild-type kinases in normal tissues limits the doses that can be used to target the mutant kinase in the tumor tissue. The key question, in light of the complex kinase networks in all cells, is which anti-targets should be avoided, in order to limit toxicity in normal tissues. Our work (*Kung et al., 2006*; *Dar et al., 2012*) and that of others (*Ahmad and Eisen, 2004*; *Wilhelm et al., 2006*) have highlighted the potent effects of multi-targeted kinase inhibitors, revealing unexpected effects when several kinases are inhibited rather than each one individually (synthetic lethal effects as well as positive epistatic effects). By understanding the synthetic lethal effects on normal cells and developing selective inhibitors which avoid even a small number of 'off-target' kinases, we believe that clinical agents with an improved therapeutic index can be developed.

One disease where this anti-target (*Dar et al., 2012*) concept is particularly needed is AML, a rapidly fatal blood cancer comprising 2% of cancer deaths in the United States in 2013 and a disproportionate number of new cases (ca. 19,000, 30%) and deaths (ca. 10,500, 44%) relative to all leukemias (*Leukemia and Lymphoma Society, 2014*). FDA-approved chemotherapy has not advanced beyond

**eLife digest** Major advances in cancer therapy have improved the treatment options for many patients. However, many cancer treatments are toxic or have severe side effects, making them difficult for patients to tolerate. One cause of these side effects is that many cancer therapies kill both normal cells and cancer cells. Developing cancer therapies that are more targeted is therefore a priority in cancer research.

Acute myeloid leukemia is a type of blood cancer that has proven difficult to treat without causing serious side effects. This cancer is very aggressive and only about 1 in 4 patients are successfully cured of their cancer. At present, physicians treat acute myeloid leukemia with chemotherapy, which kills both the cancer cells and some of the patient's healthy cells.

Many patients with acute myeloid leukemia have mutations in the gene encoding an enzyme called Fms-like tyrosine kinase 3 (FLT3). This mutation makes the enzyme permanently active, and patients with the mutation have a greater risk of their cancer recurring or death. Scientists have recently discovered that treatments that inhibit the FLT3 enzyme can be effective against cancer. However, the drugs investigated so far also interfere with the patient's ability to produce new blood cells, which can lead to infections or an inability to recover from bleeding. Therefore, no new drugs have yet been approved for general use.

Warkentin et al. suspected the reason for the adverse effects of FLT3 inhibitors is that these drugs also inhibit another enzyme necessary for blood cell production. Previous work showed that inhibiting one or the other of the enzymes still allows blood cells to be produced as normal: it is only when both are inhibited that production problems arise. Warkentin et al. therefore looked for a chemical that inhibits only the FLT3 enzyme and found one called Star 27. Tests revealed that this inhibits FLT3 and prevents the growth and spread of cancerous cells but does not impair blood cell production. Additionally, Star 27 continues to work even when mutations arise in the cancer cells that cause resistance to other FLT3 inhibitors.

The findings demonstrate that when it comes to drug development, it is sometimes as important to avoid certain molecular targets as it is to hit others. Understanding the network of enzymes that FLT3 works with could therefore help researchers to develop more effective and safer cancer treatments.

general cytotoxic agents, and no highly active therapies have been approved in over 30 years. The cure-rate for AML remains approximately 25%. Fms-like tyrosine kinase 3 (FLT3) is a receptor tyrosine kinase (RTK) regulating hematopoietic differentiation that is mutated in 30–35% of AML cases, making it the most frequently mutated gene in AML (*Mizuki et al., 2000*). 25% of AML patients present with juxtamembrane (JM) domain duplications termed i̲nternal t̲andem d̲uplications (FLT3-ITD) (*Kindler et al., 2005*). The ITD mutation is thought to render FLT3 constitutively active by disrupting an auto-inhibitory function of the JM domain. Patients with FLT3-ITD mutations have significantly increased relapse rates and shortened survival, thus illustrating a major unmet therapeutic need.

FLT3-ITD was recently validated as a therapeutic target in AML (*Smith et al., 2012*). Several targeted FLT3 tyrosine kinase inhibitors (TKIs) have been investigated; however, none have advanced beyond Phase III clinical trials (*Smith and Shah, 2013*). The FLT3 TKI AC220 (Quizartinib) (*Chao et al., 2009*) achieves substantial reduction in leukemic blasts initially in a high proportion of patients and is the most kinome-wide selective clinical candidate (*Zarrinkar et al., 2009*). However, in spite of its promising selectivity, which is remarkably confined to inhibition of the Class III/PDGFR family of RTK's, AC220-induced myelosuppression represents a major dose-limiting toxicity (*Galanis et al., 2012*; Smith and Shah). As a result, while many patients achieve clearance of bone marrow (BM) blasts, most experience incomplete recovery of normal blood counts (CRi) and remain at risk of complications such as life-threatening infection or bleeding.

The Class III family of RTKs (comprising FLT3, KIT, CSF1R, PDGFRα, and PDGFRβ) are important regulators of normal hematopoiesis. In 1995, Lemischka and coworkers showed that mice lacking function of either Flt3 or Kit maintained overall normal populations (*Mackarehtschian et al., 1995*). However, mice lacking both Flt3 and Kit function had a dramatic reduction of hematopoietic cell numbers, ca. 15-fold white cell depletion, reduction of lymphoid progenitors, and postnatal lethality. We

propose that KIT is an anti-target (**Dar et al., 2012**) in the context of pharmacologic inhibition of FLT3. Thus, normal mature hematopoietic populations can be maintained in the context of either Flt3 or Kit inhibition alone but not dual Flt3/Kit inhibition (**Bershtein et al., 2006**).

This synthetic lethal toxicity relationship between FLT3 and KIT for maintaining normal hematopoietic populations may explain the adverse side effects of the current kinase targeted drugs in clinical development. In a recent single agent Phase II trial, PKC412 failed to achieve a single complete remission (CR). When combined with cytotoxic agents PKC412 showed some promise, achieving a 25% CR rate, but responses were primarily incomplete recovery of peripheral blood counts (CRi, 20%) with over 90% of patients developing grade 3/4 myelosuppression (**Strati et al., 2014**). While AC220 monotherapy impressively demonstrated a 50% CR rate in a Phase II trial, these consisted primarily of CRi (45%) with few real CRs with complete recovery of blood counts (**Cortes et al., 2013**), correlating with the similar potency of these agents for both FLT3 and KIT. A recent study showed increased selectivity of the clinical agent crenolanib for FLT3 over KIT and reinforced the correlation between target inhibition, and anti-target avoidance (**Dar et al., 2012**), which lead to lowered toxicity towards normal hematopoiesis (**Galanis et al., 2014**). However, the potency of crenolanib for KIT remains too high ($IC_{50}$ = 67 nM for p-KIT inhibition in TF-1 cells; 65% inhibition at 100 nM, in vitro) (**Galanis et al., 2014**). This is likely insufficient to fully minimize clinically relevant myelosuppression, as a recent interim analysis reported only a 17% (3/18 patients) composite CR rate in AML patients, with 2/3 of these responders achieving only CRi (**Collins et al., 2014**). These findings highlight the need for new clinical candidates that better minimize KIT and other Class III RTK inhibition.

While avoiding inhibition of the presumed anti-target, KIT, is one chemical challenge toward inhibitor design, the emergence of on-target resistance is another clinical challenge. We (**Smith et al., 2012**) and others (**Wodicka et al., 2010**) have identified the acquisition of secondary FLT3 kinase domain (KD) mutations that cause drug resistance as another limitation of current clinically active FLT3 inhibitors. Mutations at the activation loop residue D835 are particularly clinically problematic. These mutations are proposed to bias the kinase toward the constitutively *active* conformation by disrupting a hydrogen bond from D835 to S838, and thus limit the efficacy of Type II inhibitors such as AC220. We have recently proposed that a Type I inhibitor, which binds to the active kinase conformation, would circumvent these mutations that confer resistance to AC220 (**Smith et al., 2012**). New small molecule therapies have been reported to bypass these particular mutations, including crenolanib (**Galanis et al., 2014**), a Type I inhibitor (**Lee et al., 2014**; **Smith et al., 2014**), but the CR rate of crenolanib remains modest (**Collins et al., 2014**). Moreover, it is likely that a repertoire of drugs will be necessary to combat emerging resistance.

We propose herein a solution to the FLT3/KIT selectivity problem designed to avoid myelosuppression and also retain potency against drug-resistant mutations. The staurosporine scaffold has been utilized pharmacologically for 30 years, and staurosporine analogs have been proven to be potent FLT3 inhibitors (PKC412, CEP701) (**Strati et al., 2014**), though clinical activity of these compounds has been modest, perhaps caused by lack of potent FLT3 inhibition due to dose-limiting toxicity in vivo. The lactam ring C7 position remains virtually unexplored for modulating selectivity (**Wood et al., 1999**; **Bishop et al., 2000**; **Heidel et al., 2005**). We recently reported that C7-substituted staurosporine analogs, we term 'staralogs', are potent and selective inhibitors of engineered analog-sensitive (AS) kinases (**Lopez et al., 2013**). For example, when C7 ($R_1$) equals isobutyl (Star 12), AS Src kinase is potently inhibited but WT kinases remain unaffected. However, we also observed that Star 12, in a panel of 319 kinases, weakly inhibits only one WT kinase, FLT3 (57% inhibition at 1 μM; KIT, CSF1R, PDGFRα/β all inhibited <10%). Thus, the C7-alkyl group of Star 12 may allow for weak but selective inhibition of FLT3 over the anti-target KIT, which contributes to myelosuppression when FLT3 is also inhibited. Although substitution of an isobutyl group at C7 reduced potency, we hypothesized that combining the FLT3/KIT selectivity of Star 12 and the potency features of PKC412 would generate an optimal FLT3 inhibitor also capable of targeting emerging mutations (**Figure 1A**).

## Results

### Discovery and in vitro structure activity relationships (SAR) towards Star 27, a FLT3/KIT selective drug

In an effort to identify an optimal C7 substituent that retains selectivity away from KIT while enhancing potency for FLT3, we synthesized and tested a panel of C7-substituted staralogs (**Figure 1B**). The

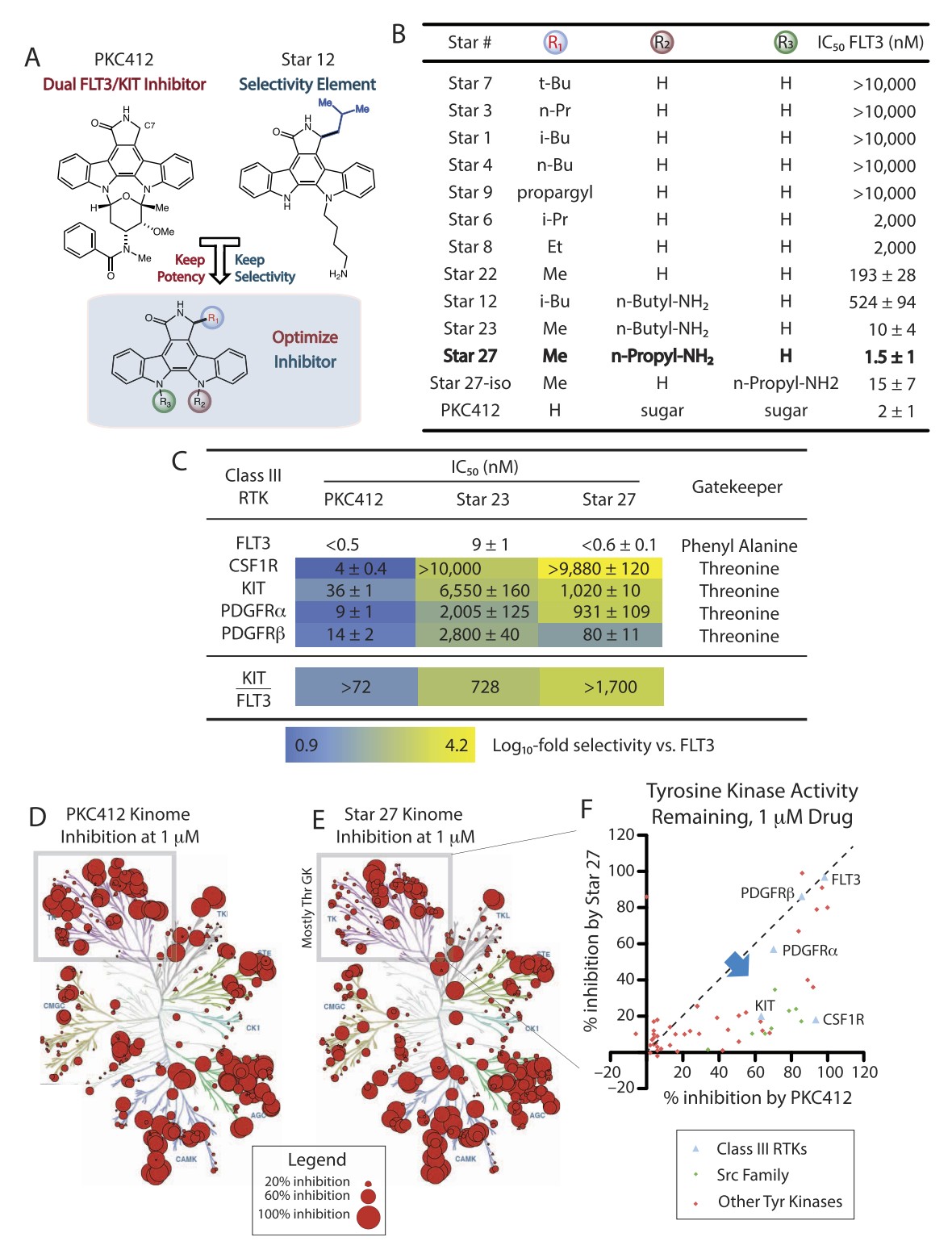

**Figure 1**. Purified kinase assays comparing SAR of staralogs. (**A**) PKC412 (N-benzoylstaurosporine) potently inhibits all Class III/PDGFR family hemato-poietic stem cell kinases. Staurosporine analog (staralog) Star 12, with large alkyl C7 (R1) substituent is a weak inhibitor of FLT3 but suggested more selectivity for Class III RTKs. (**B**) Table shows purified kinase in vitro assays testing SAR showing optimal potency at R1 = methyl; rigid potency window found with Star 27 with n-propyl-NH2 at R2 but not R3. Star ## ordered by increasing potency and numbering consistent with prior usage (**Lopez et al., 2013**). Error ranges represent standard error of the mean (SEM) and are the result of at least three independent measurements, each in triplicate.

*Figure 1. Continued on next page*

*Figure 1. Continued*

(**C**) $IC_{50}$ values of PKC412 and lead compounds Star 23 and 27 against the hematopoietically relevant five Class III RTKs showing the optimal selectivity achieved for Star 27. $Log_{10}$-scale heat map highlights $IC_{50}$ ratios relative to FLT3, indicating progression from PKC412 having lower selectivity towards Star 27 having high selectivity. (**D**) Dendrogram showing single point inhibition for 319 kinases for PKC412. Each value represents the average of two experiments ± SEM (performed by RBC). (**E**) Dendrogram showing single point inhibition for 319 kinases for Star 27. (**F**) Graph of Tyr kinases. y-axis = potency by single point for Star 27 as a function of the corresponding potency with PKC412.

C7 substituent points toward the gatekeeper (GK) residue of the kinase (*Lopez et al., 2013*), and FLT3 and KIT possess Phe and Thr gatekeepers, respectively. We chose an unbiased selection of groups easily derived from the chiral pool, and tested these derivatives in a radiolabeled purified kinase assay, finding that the alanine-derived methyl group provided the most potent inhibition. Synthesis of C7 analogs was further expedited by testing the fully aglyconic derivatives (see *Figure 1B*, entries 1–8). Replacement of the sugar moiety of PKC412 with a pendant amine has proven to be a means to reduce the chemical synthetic burden of carbohydrate synthesis while simultaneously maintaining potency. Attachment of *n*-propyl amine to mimic the *N*-methyl amide of PKC412 produced comparable values (*Figure 1B*, Star 27: 1.5 nM; PKC412: 2 nM). Interestingly, both Star 27's one carbon homolog (Star 23) and regioisomer (Star 27-*iso*) reduced activity, further indicating a well-defined structure activity relationship (*Figure 1B*).

We next evaluated our lead inhibitors to show the role of C7 modification on the ratio of KIT/FLT3 $IC_{50}$s. While PKC412 exhibited potent inhibition of all five Class III RTKs (*Figure 1C*, $IC_{50}$ = <0.5–36 nM), Star 23 showed improved differentiation between FLT3 and other Class III RTKs ($IC_{50}$ = 9 – > 10,000 nM), and Star 27 proved to have the best selectivity ($IC_{50}$ = <0.6–9880 nM). This selectivity increases from (a) KIT/FLT3 ratio of >72 for PKC412 to >1700 for Star 27 and (b) CSF1R/FLT3 ratio of 8 for PKC412 to >16,500 for Star 27 (*Figure 1C*). Correlation is seen with inhibition of kinases containing Phe GKs (e.g. FLT3) and avoidance of those containing Thr GKs (e.g. KIT) for C7-Me containing staralogs.

We then profiled Star 27 against 319 purified kinases using the same assay conditions used previously for PKC412. We chose to screen for additional targets at 1 μM, or 1000-fold above the $IC_{50}$ value for the desired target, which captures the maximum number of targets of each inhibitor, dendrograms showing in *Figure 1D* (PKC412) and *Figure 1E* (Star 27). At this high concentration, the two drugs inhibit many of the same targets. Yet the two drugs show important differences in the tyrosine kinase family. The percent inhibition of 54 Tyr kinases by Star 27 with respect to their percent inhibition by PKC412 is shown in *Figure 1F*. This plot indicates a substantial shift towards inhibition by PKC412 but not Star 27 and correlates with a high number of Thr GK residues (42 Thr GKs for Tyr kinases vs 34 Thr GKs in non-Tyr kinases). This shift is particularly striking for the entire Src family (*Figure 1D*, pink triangles), all members of which possess Thr GKs. Conversely, Tyr kinases that possess Phe GKs mostly display high and equipotent inhibition by both Star 27 and PKC412 (including FLT3, TRKA, TRKB, and TRKC).

## Star 27 demonstrates increased selectivity for FLT3 over KIT in cellular models

We compared PKC412 to Star 27 for the ability of each to inhibit cell proliferation of human cell lines addicted to FLT3 or KIT. We employed Molm14 and MV4;11 cells, harboring the FLT3-ITD mutation (hetero- and homozygous, respectively); HMC1.1 cells (dependent on KIT V560G); and K562 cells (which express the BCR-ABL fusion kinase) as a control for non-KIT-related toxicity. Sorafenib, AC220, and ponatinib all displayed equipotent inhibition of FLT3 and KIT-driven cell lines (see *Figure 2A* and reference values [*de Jong and Zon, 2005*; *Guo et al., 2007*; *Kampa-Shittenhelm et al., 2013*]). PKC412 increases this selectivity 22-fold (HMC1.1/MV4;11). Encouragingly, Star 27 inhibited proliferation of the FLT3-ITD+ cells while not affecting HMC1.1 proliferation leading to a selectivity window of >122-fold (HMC1.1/MV4;11).

We next tested if this inhibition of proliferation was manifested in triggering apoptotic cell death. Analysis of caspase-3 activation revealed a similar induction of apoptosis in a dose–response manner between PKC412 and Star 27 in Molm14 cells (*Figure 2B*). Conversely, PKC412-induced apoptosis in the KIT mutant HMC1.1 cells as predicted while minimal apoptosis was observed with Star 27, further highlighting the selectivity of Star 27 for FLT3 over KIT (*Figure 2C*).

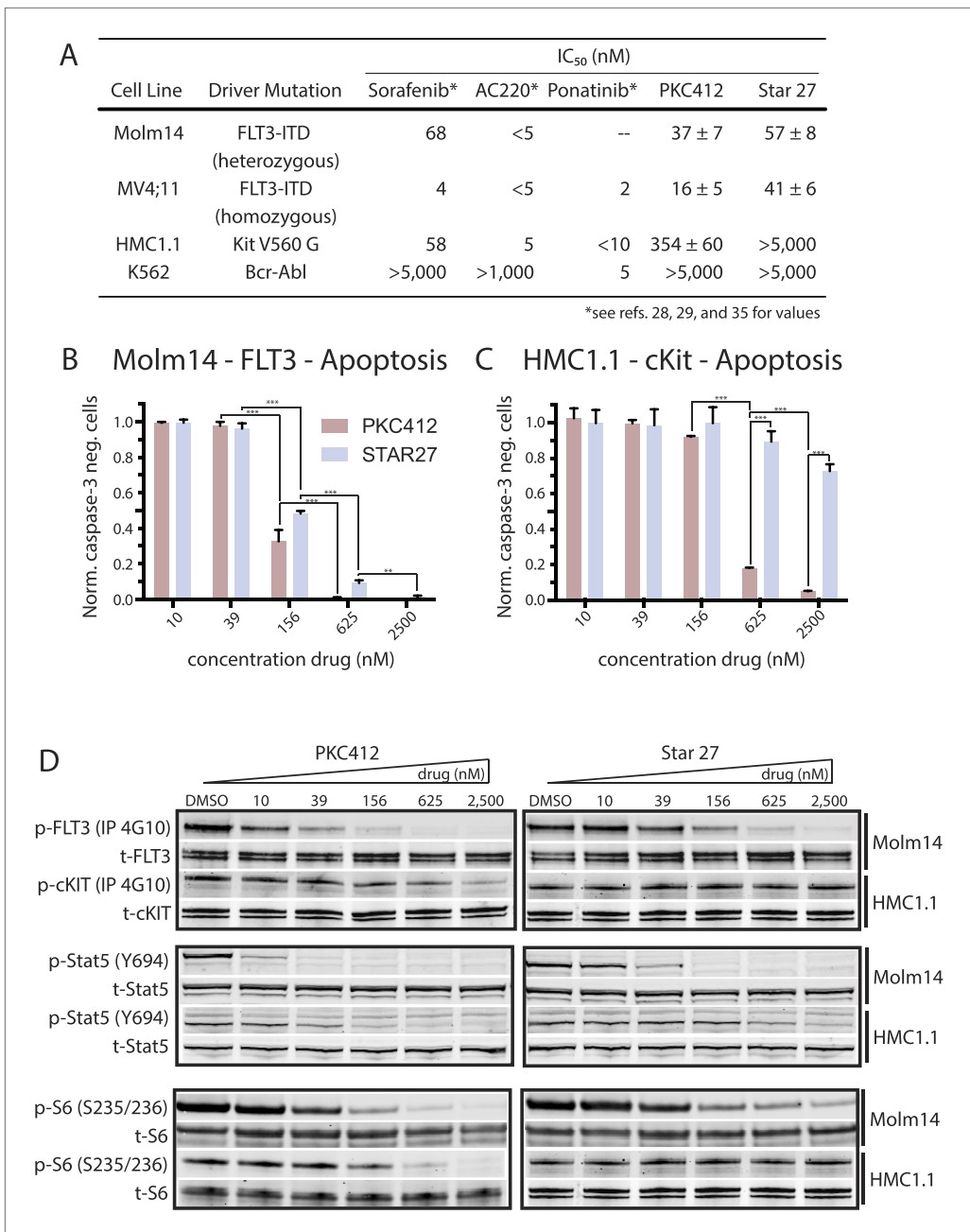

**Figure 2**. Cellular proliferation, apoptosis, and biochemical validation of Star 27's potency against FLT3 mutants, but not in the KIT context. (**A**) Table of cellular IC$_{50}$ values for leading experimental clinical therapies (Sorafenib, AC220, Ponatinib, and PKC412) and Star 27 against a panel of AML-relevant human-derived (MV4;11 and molm14) cell lines and toxicity controls (HMC1.1 and K562) (adapted* from *Guo et al., 2007*; *Kampa-Shittenhelm et al., 2013*; *Smith et al., 2013*). Star 27 shows a similar order of magnitude potency against Molm14 and MV4;11 compared to PKC412. Relevant therapeutic windows between PKC412 and Star 27: an over fivefold increase in selectivity for HMC1.1/MV4;11 cells and an 11-fold increase for HMC1.1/Molm14 is observed for Star 27 over PKC412. Replicates shown are the result of at least three attempts, each in triplicate, and error ranges represent the standard error of the mean. (**B**) Normalized caspase-3 negative cells plotted against escalating drug dosage. Star 27 shows a similar degree of apoptosis to PKC412 in Molm14 cells. (**C**) In KIT-addicted HMC1.1 cells PKC412 exhibits potent toxicity while Star 27 shows very little up to 2,500 nM. Results performed in triplicate with two biological replicates. **p < 0.01; ***p < 0.001. (**D**) PKC412 and Star 27 have a similar degree of inhibition against p-FLT3 in Molm14 cells as well as a similar degree for p-STAT5 and p-S6 (for p-ERK, p-AKT, and p-MEK data not shown). In the HMC1.1 cells, by contrast, PKC412 and Star 27 show a larger difference in p-KIT inhibition.

Given the similarity in potency between PKC412 and Star 27 towards FLT3, we next tested both compounds for their ability to inhibit FLT3 autophosphorylation as well as downstream signaling in Molm14 cells. We observe a similar inhibition of p-FLT3 inhibition between the two drugs and downstream phosphorylation among three canonical signaling arms was uniform between PKC412 and Star 27 (JAK/STAT, RAS/ERK, and mTOR/AKT/S6; *Figure 2D*). In contrast to similar phospho-signaling inhibitory profiles in Molm14 cells, PKC412 inhibited p-KIT and downstream signaling to a greater degree in HMC1.1 cells compared with Star 27 (*Figure 2D*).

## PKC412 but not Star 27 exhibited myeloid toxicity in an ex vivo model of human hematopoiesis while both inhibited primary AML blast colony formation

We next asked if Star 27's FLT3/KIT selectivity could maintain efficacy and reduce toxicity in primary patient-derived contexts. Colony-forming assays are useful for this application because they better represent the microenvironment of the BM niche than traditional cell proliferation of primary human cells (*Miller and Lai, 2005*). Testing both PKC412 and Star 27 demonstrated their ability to inhibit the growth of primary patient-derived FLT3-ITD+ blasts in colony-forming assays, with both demonstrating >80% inhibition of colony formation at 1000 nM (*Figure 3A,B*). Because of the importance of maintaining potency against KD mutations in addition to those of the JM domain, we also tested the ability of both drugs to inhibit the colony-forming ability of primary patient blasts containing a FLT3-D835 mutation (ITD−; point mutational status undetermined). Star 27 inhibited about 60% of colonies at 1000 nM while PKC inhibited about 80% (*Figure 3C*).

Alternatively, we tested PKC412 and Star 27 for their effects on hematopoietic colony formation of normal BM and stimulated peripheral blood (SPB) samples. One of the four donor samples is shown in *Figure 3D*. *Figure 3E,F* shows quantification of colony counts for both erythroid (or burst)-forming units (BFUs) and colony-forming units (CFUs). PKC412 demonstrated dose-dependent inhibition of colony formation from normal hematopoetic progenitors while Star 27 demonstrated no significant effect up to a concentration of 1,000 nM (*Figure 3E,F*).

## In vivo zebra fish model shows Star 27 to be non-myelosuppressive, PKC412 to be myelosuppressive, and both drugs to be efficacious against FLT3-ITD AML

Having validated Star 27's biochemical potency and lack of myelosuppression, we next tested its effects in vivo. We chose the established model, *Danio rerio* (zebrafish) for studying hematopoiesis (*Davidson and Zon, 2004*; *de Jong and Zon, 2005*). In this model, embryos injected with FLT3-ITD and treated with AC220 have been shown to recapitulate knock-down of FLT3-ITD in clinical studies (*He et al., 2014*). This model also provides for robust and sensitive measures of in vivo myelopoiesis (*He et al., 2014*).

We treated embryos with Star 27 up to 10 μM and studied the effects on WT morphology at 3 days post fertilization (dpf), finding no change in heart and tail morphology (tail length and lack of curvature, see *Figure 4A–D*). In contrast, PKC412 showed substantial morphological defects to the heart at 1 μM vs vehicle (*Figure 4E–I*), tail length and tail curvature at 1 μM and 2.5 μM (see *Figure 4E,G*, respectively).

We next examined the effect of Star 27 and PKC412 on WT granulopoiesis as a measure of KIT-related myelopoiesis/myelosuppression measured by myeloperoxidase (mpo)-stained in situ hybridization in the posterior blood island (PBI). Representative cross sections for the PBI for DMSO, Star 27, and PKC412 are shown (*Figure 4J,K,L*, respectively). Quantification of mpo dots in Star 27-treated embryos shows a non-significant difference between 10 μM and DMSO treatments of 30 hr post fertilization (hpf, *Figure 4J,K,M*). In contrast, PKC412 showed statistically significant suppression of myelopoiesis at 10 μM (*Figure 4M*).

Efficacy in the AML FLT3-ITD context is quantified by binning the FLT3-ITD blast phenotype into Normal, Intermediate, and Severe states (*Figure 4N–P*, respectively) (*He et al., 2014*). Star 27 did have efficacy in preventing the spread of neutrophilic blasts in FLT3-ITD-transduced embryos (*Figure 4Q*) similar to that previously seen with AC220 (*He et al., 2014*). Similar to Star 27, PKC412 also showed an ability to limit the spread of FLT3-ITD-injected leukemic blasts (*Figure 4Q*).

## Calculation predicts selectivity for Star 27 based on electronics and may be general beyond Class III RTKs

The structural basis of selectivity of Star 27 for FLT3 over other Class III RTKs was computationally investigated, revealing two features of staralog binding that may explain Star 27's selectivity for FLT3

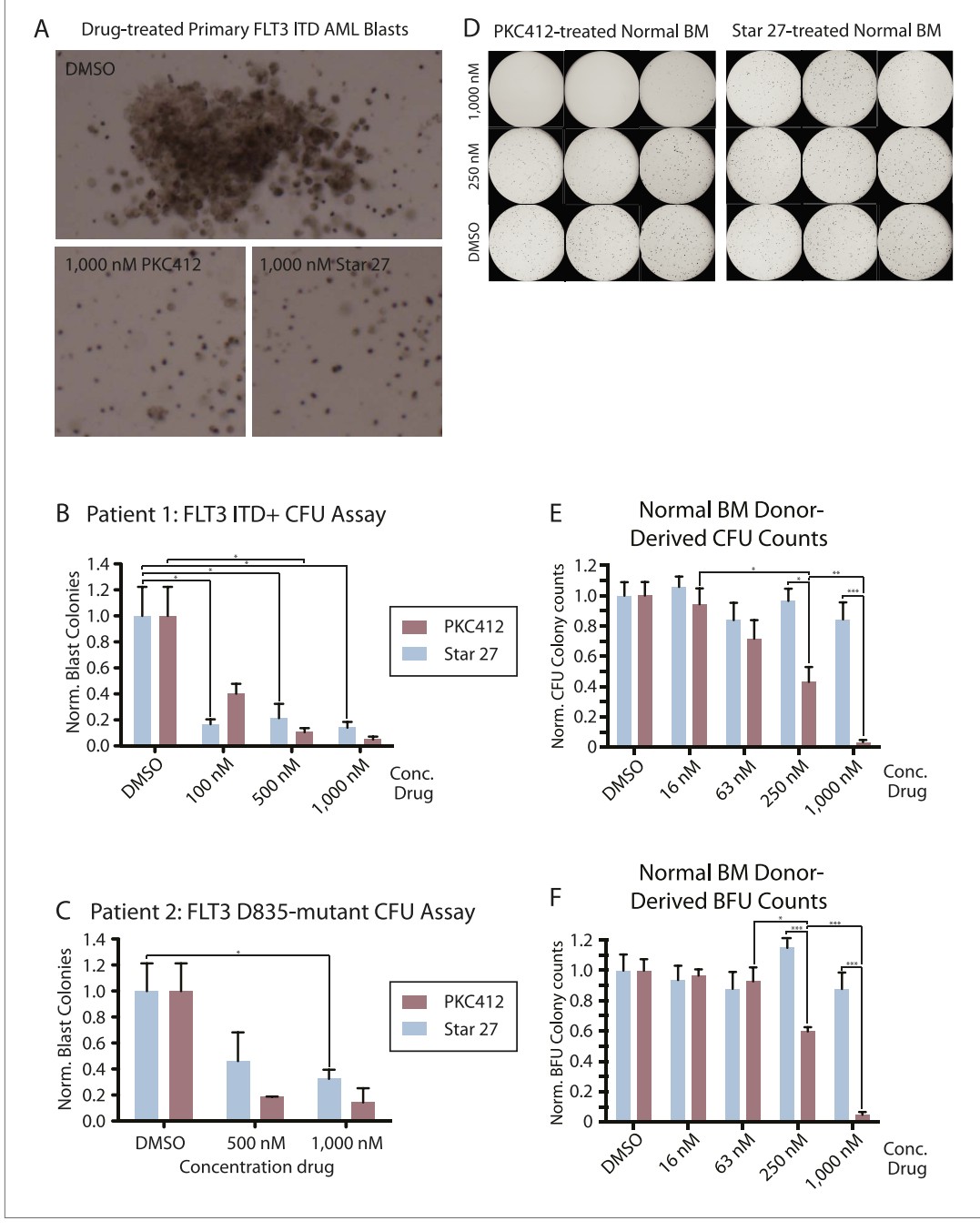

**Figure 3**. Colony forming assays comparing effect of Star 27 and PKC412 on normal donor-derived stimulated peripheral blood (SPB), normal bone marrow (BM) growth, and malignant blast reduction. (**A**, **B**, **C**) Colony forming assays comparing effects of Star 27 and PKC412 on primary patient AML circulating blast growth in methylcellulose. Colonies consist primarily of leukemic blasts of various hematopoietic identities. BFU colonies are not shown. Colony numbers scored individually for each replicate 3 cm plate. (**A**) Images showing leukemic blast morphology in DMSO-treated plates and representative images in 1,000 nM-treated plates for both drugs as indicated. (**B**) Primary patient FLT3-ITD+ AML circulating blasts. Raw colony numbers range from ca. 20 in PKC412 or Star 27-treated cells at 1,000 nM to ca. 1,200 for CFUs in DMSO-treated plates. Concentrations shown were applied in triplicate to each concentration for two different cell densities, the larger (2.5 × 10^5 cells/ml) showing adequate cell growth and colony formation for statistically significant counts, the averages of those trails then calculated for standard error of the mean. *p < 0.05. (**C**) Primary patient FLT3 D835Y AML circulating blasts. Raw colony numbers range from ca. 25 for PKC412 (or 50 for Star 27) at 1,000 nM to ca. 120 for DMSO-treated plates. Conditions

*Figure 3. Continued on next page*

*Figure 3. Continued*

repeated in duplicate or higher replicates. Colonies scored individually for the highest cell density tested ($2.5 \times 10^5$ cells/ml). Similar to primary ITD patient samples, colonies consist mostly of poly-hematopoietic leukemic blasts. *$p < 0.05$. (**D**, **E**, **F**) Normal BM and SPB colony data. (**D**) Images of colonies grown and derived from normal BM (images of SPB not shown) in methylcellulose. Raw colony numbers for BM range from zero in most PKC412-treated replicates at 1,000 nM to ca. 400 for BFUs and ca. 300 for CFUs in all Star 27-treated conditions. Non-magnified differences are particularly noticeable between the 1,000 nM and 250 nM dosages. (**E** and **F**) Concentrations shown were applied in triplicate to three normal SPB and one normal BM donors, the averages of those trials then calculated for standard error of the mean (full data not shown). Colony counts for DMSO ranged from the low 100's to 400's. Graphs show Star 27 having no effect on hematopoiesis up to 1,000 nM while PKC412 eliminates most normal hematopoietic colony forming potential at 1 mM on colony forming units (CFU) and blood forming units (BFU), respectively. CFU-GEMM colonies were counted as one CFU plus one BFU. *$p < 0.05$; ***$p < 0.001$.

over KIT, vs PKC412's equipotency towards both kinases. (a) Electronics: we calculate that PKC412 maintains a partial positive charge at its lactam C7 position, matching the partial negative charge of KIT's Thr GK (*Figure 5A–C*), while Star 27's C7 methyl renders its lactam relatively neutral (*Figure 5D–F*), reducing this affinity interaction. (b) Sterics: staralogs with no C7 substitution maintain a flat lactam for binding to the hinge region, adjacent to the GK. Based on crystallographic data from KIT's active conformation (*Gajiwala et al., 2009*), a simple steric argument may explain why Star 27's C7 methyl group prevents binding to KIT's restricted ATP binding pocket while PKC412 potently binds (data not shown). Conversely, FLT3's active site is hypothetically large enough to accommodate the 17.4 Å³ van der Waal's volume of methyl (vs. 1.17 Å³ for H). Taken together both arguments may explain the greater affinity of PKC412 for both kinases and of Star 27 for FLT3 but not KIT (calculated using MOE ver2013.0801). This model may account for the selectivity seen for Star 27 for kinases bearing Phe GKs (FLT3, TRK, etc) and away from those bearing Thr GKs (KIT, CSF1R, Src family, other Tyr kinases, see *Figure 1D*).

## Star 27 shows uniform potency against drug-resistant mutant cell line proliferation

Mutations that confer drug resistance to existing Type II FLT3 inhibitors AC220 and sorafenib commonly prevent the kinase from efficiently adopting an inactive conformation required for drug binding. Type I FLT3 inhibitors, such as PKC412 and Star 27, are predicted to be less vulnerable to such mutations. We tested both inhibitors against a panel of 17 cell lines containing different FLT3 TKI mutations with varying resistance to the leading clinical candidate therapies, sorafenib, ponatinib, and AC220 (*Figure 6A*). IC$_{50}$ values are presented as log$_{10}$-fold resistance in a color-coded heat map to highlight the range of offset. Using published IC$_{50}$ values for sorafenib, AC220, and ponatinib for comparison (*Guo et al., 2007*; *Kampa-Shittenhelm et al., 2013*; *Smith et al., 2013*), we observed equipotent values for PKC412 against most KD point mutants. Intriguingly, Star 27 maintained similar potency against most mutants compared to FLT3-ITD alone.

## Crenolanib is a potent inhibitor of p-KIT and normal BM colony formation

Crenolanib has been reported to inhibit p-KIT in TF-1 cells (*Galanis et al., 2014*) and HMC1.2 cells (*Smith et al., 2014*) at 67 nM and <100 nM IC$_{50}$'s, respectively. In order to directly compare our results with PKC412 and Star 27 (*Figure 2D*), we tested crenolanib's biochemical inhibition of p-KIT in HMC1.1 cells. *Figure 6B* shows that crenolanib inhibits p-KIT in these cells at an IC$_{50}$ of 12 nM. This potency against KIT anti-target is propagated down to inhibition of p-AKT and p-S6 kinases. Crenolanib's potency against p-FLT3 has been reported to be 5 nM (*Galanis et al., 2014*).

Crenolanib has been reported to have only modest inhibition of normal BM colony formation of CFUs and no inhibition of BFUs up to 200 nM drug treatment (*Galanis et al., 2014*). However, the lack of both higher drug dosing (>200 nM) and statistically comparable variation prompted us to retest crenolanib in the same colony forming assay up to 1,000 nM, to compare to PKC412 and Star 27 (*Figure 3D–F*). We observe crenolanib to have a more substantial dampening of normal colony growth of both CFU and BFU subtypes in normal BM (see *Figure 6C*).

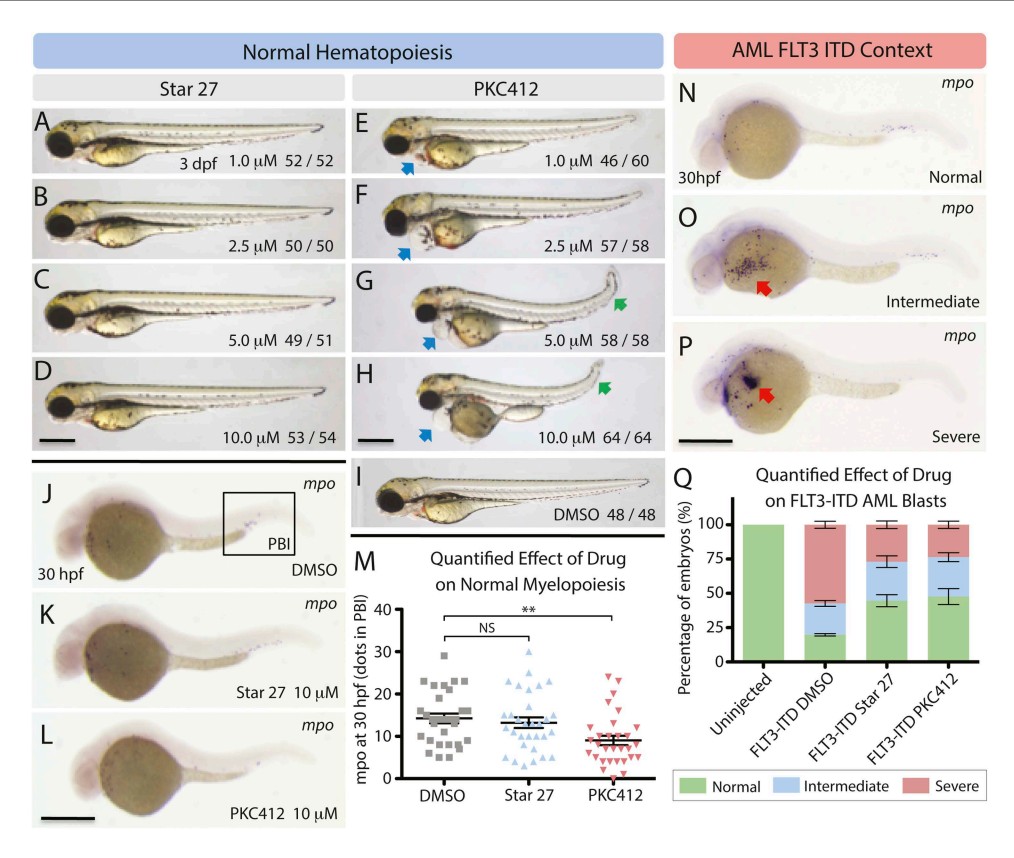

**Figure 4**. Effects of Star 27 on *D. rario* (zebrafish) WT morphology and myelopoiesis, and FLT3-ITD AML context. Effect of Star 27 on zebrafish normal myelopoiesis and the FLT3/ITD-induced myeloid cells expansion. (**A**–**D**) The effect of Star 27 on zebrafish embryonic development at 3 dpf, showing no noticeable morphological defects up to 10 μM. (**E**–**I**) The effect of PKC412 on zebrafish embryonic development at 3 dpf, showing substantial pericardial defects beginning at 500 nM (data not shown), and tail curvature and length defects beginning at 1 μM and 2.5 μM, respectively vs. vehicle. (**J**, **K**, **M**) The effect of 10 μM Star 27 treatment on mpo+ myeloid cells development in the posterior blood island (PBI) at 30 hpf, showing no statistically significant change. (**J**, **L**, **M**) The effect of 10 μM PKC412 treatment on mpo+ myeloid cells development in the PBI at 30 hpf, showing a statistically significant granulogenesis/myelosuppression. (**N**–**P**) Three categories of *mpo* transcription (**N**, normal; **O**, intermediate; **P**, severe) were defined based on the WISH results (three experiments). (**Q**) The rescue effect of 10 μM Star 27 treatment on FLT3/ITD-induced mpo+ myeloid cells expansion at 30 hpf, showing rescue of normal phenotype approaching that seen for AC220 (*He et al., 2014*). Scale bar equals 500 μm. Blue arrows indicate the pericardial edema, green arrows indicate tail shortening and curving, and red arrows indicate the FLT3-ITD AML mpo+ myeloid cells expansion. PKC412 treatment on FLT3/ITD-induced mpo+ myeloid cells expansion at 30 hpf, showing an efficaciousness similar to Star 27, consistent with *Figures 1–3*. For all experiments: Zebrafish embryos were collected and kept in standard E3 medium at 28°C. Different concentrations of either drug were added to the E3 medium from 6 hr post fertilization (hpf) to 3 days post fertilization (dpf). Embryos treated with DMSO or 10 μM drug from 6 to 30 hpf were collected for *mpo* whole mount in situ hybridization (WISH) analysis. 80 ng plasmid DNA containing FLT3/ITD sequence was microinjected into one-cell stage embryos, and the uninjected embryos were used as control. FLT3/ITD-injected embryos were treated with DMSO or 10 μM drug from 6 to 30 hpf. Embryos were collected at 30 hpf for *mpo* WISH analysis. **p <0.01.

# Discussion

The aggressive nature of FLT3-driven AML and the lack of effective and well-tolerated targeted therapy represent a major unmet therapeutic need. Currently, the most promising clinical targeting agents cause myelosuppression in most patients regardless of remission status. As kinase inhibitor-based therapies evolve, refined selectivity against particular off-targets (so-called anti-targets [*Dar et al., 2012*]) will drive chemical design (*Ciceri et al., 2014*). Avoiding specific anti-targets will hopefully lead

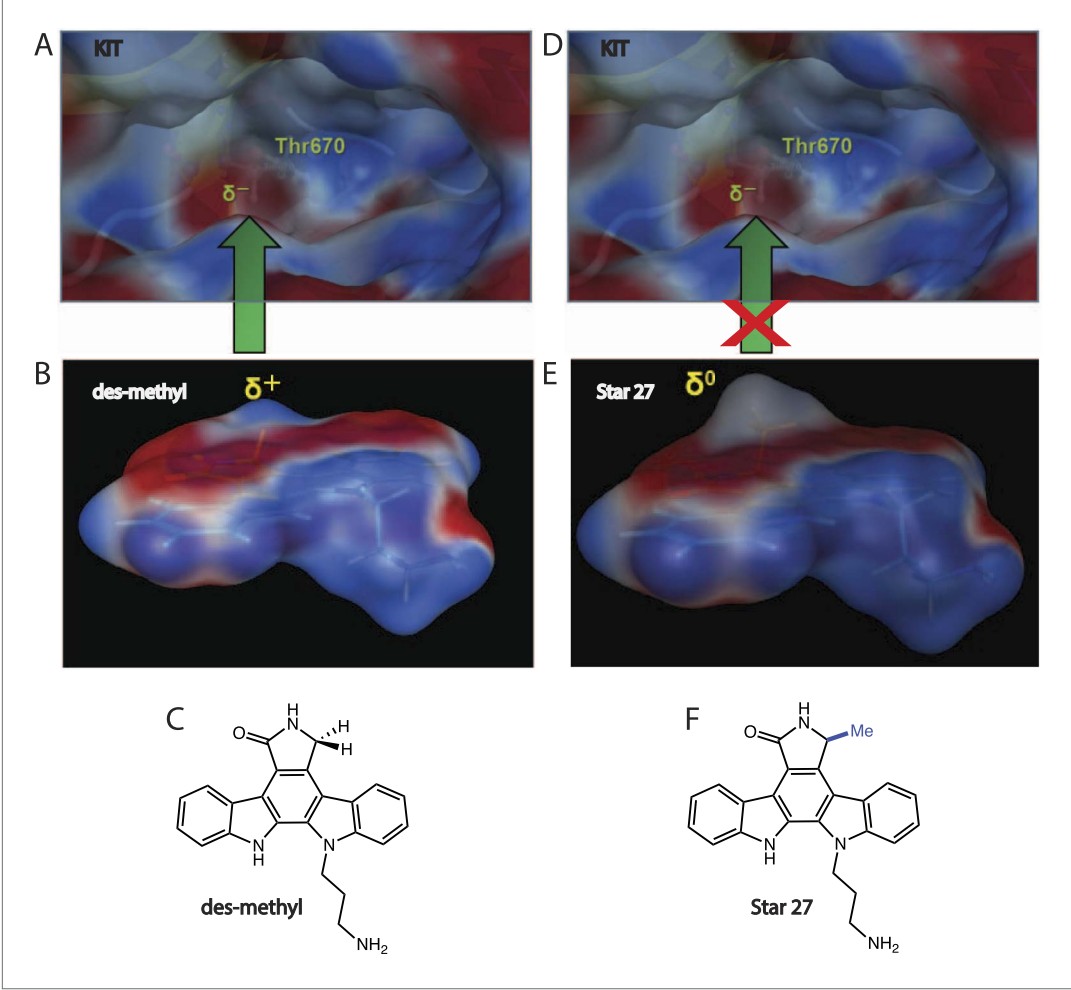

**Figure 5**. Electronic model of selectivity. Star 27-des methyl (PKC412-relevant) and Star 27 are depicted in 3D with electron potential mapping (EPM) to highlight the difference of a single methyl group. (**A–C**) Star 27-des methyl maintains a partial positive charge at the C7 position which matches the partial negative charge on the Thr 670 gatekeeper of KIT, potentially explaining PKC412's relative affinity for KIT. (**D–F**) Star 27 maintains a neutral charged surface at C7 due to the presence of the methyl group, potentially explaining its lack of binding to KIT.

to a reduction in dose-limiting toxicities that will allow more complete inhibition of the oncogenic drivers of disease.

A former targeted clinical candidate, tandutinib (MLN518), may further demonstrate directly the effects of the FLT3/KIT synthetic lethal toxicity model (*Griswold et al., 2004*). MLN518 was indicated to have similar $IC_{50}$s for FLT3 and KIT phosphorylation (220 and 170 nM, respectively). While MLN518 showed promise with reducing FLT3-ITD+ blasts in primary patient-derived samples, exposure of the drug to the BM of healthy individuals at a dose necessary for primary FLT3-ITD+ CFU reduction, led to severe depletion of healthy hematopoietic colony formation at 1 µM.

The most direct evidence exists for FLT3/KIT as a synthetic lethal toxic pair, however, other Class III RTKs may exhibit synthetic lethal effects in terms of dose-limiting toxicity when treating FLT3-driven disease. Another Class III RTK, CSF1R, has been shown via the knockout 'toothless' rat model (tl = csf1r$^{-/-}$) to be important to osteoclastogenesis (*Chen et al., 2011*), colon development (*Huynh et al., 2013*), and for normal macrophage and dendritic cell (DC) production and maintenance (*Stanley et al., 1997*). Perhaps most importantly, tl rats have been reported with a 32% platelet loss, and this thrombocytopenia was not reversed by BM transplant (*Thiede et al., 1996*). Star 27 exhibits a large selectivity window with a >16,500/1 CSF1R/FLT3 $IC_{50}$ ratio (*Figure 1C*), which may provide a clinical benefit. However, more work is needed to determine the synthetic lethal toxicity of CSF1R

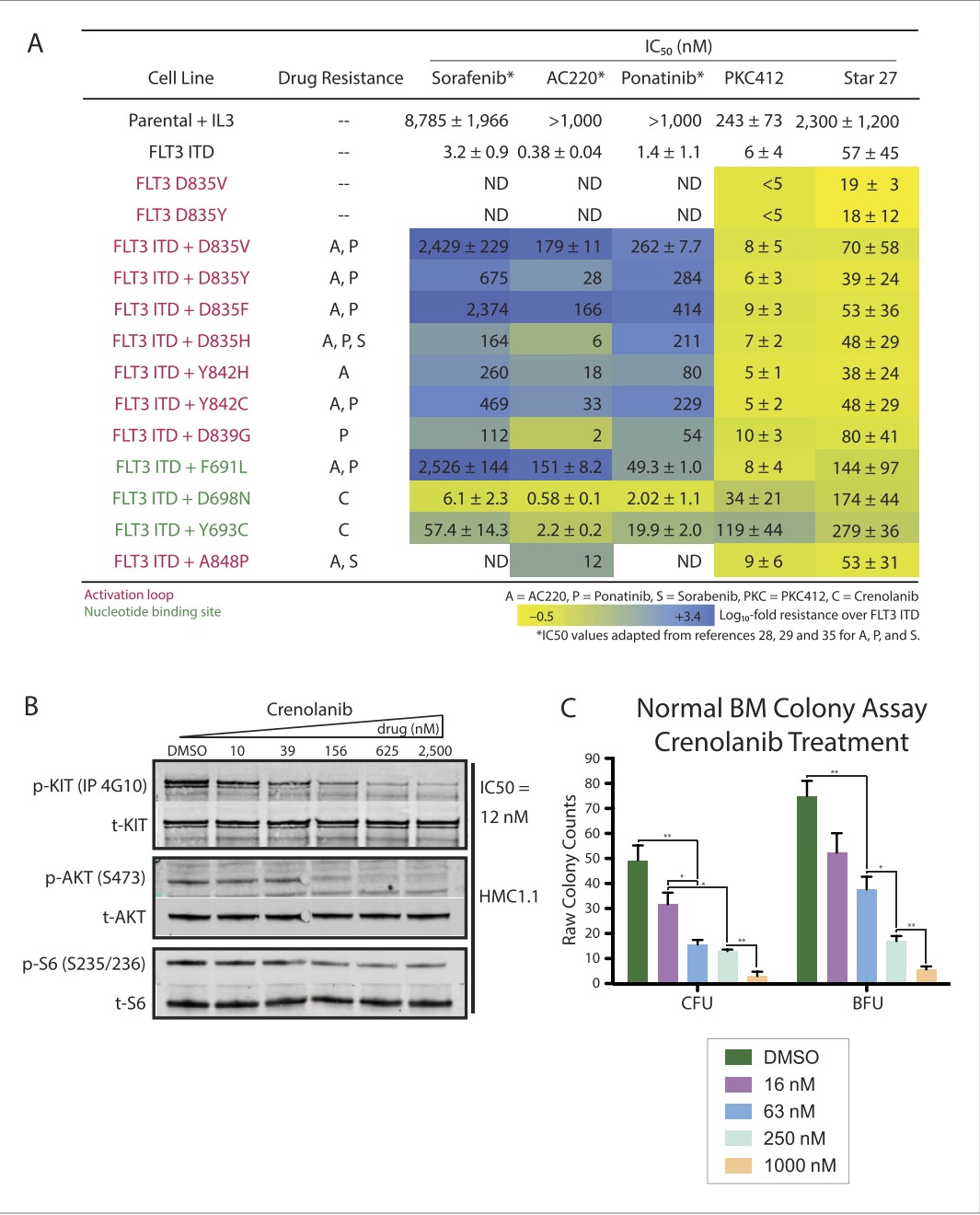

**Figure 6**. Heatmap of cellular EC$_{50}$ values for leading clinical therapies (S = Sorafenib, A = AC220, P = Ponatinib, PKC412) and Star 27 against a panel of AML-relevant human- and mouse-derived drug-resistant cell lines. (**A**) (adapted from *Guo et al., 2007*; *Kampa-Shittenhelm et al., 2013*; *Smith et al., 2013*). Log$_{10}$-scale fold resistance between Ba/F3 FLT3 ITD and drug-induced mutations (ITD + KD double mutant) shown via heat map. Point mutations corresponding to each drug-relevant resistance shown. Star 27 maintains potency between FLT3 ITD and resistance mutants comparable to PKC412. Mutations derived independently from both saturating mutagenesis and patient-derived samples [*Smith et al., 2012*]. Replicates shown are the result of at least two or three attempts, each in triplicate, and error ranges represent the standard error of the mean. (**B** and **C**) Crenolanib's effect on p-KIT inhibition and of colony growth in normal BM. (**B**) Crenolanib inhibits p-KIT in HMC1.1 cells at 12 nM IC$_{50}$. This level of inhibition translates to downstream kinases, with inhibition of p-AKT (S473) and p-S6 (S235/S236). (**C**) Crenolanib potently inhibits normal BM colony formation at <63 nM and ca. 63 nM IC$_{50}$'s of CFU and BFU colonies, respectively.

since other reports suggest an opposite and more complicated role for it in hematopoiesis (*Dai et al., 2002*).

Our focus on the staurasporine scaffold maintains the beneficial aspects of PKC412's potency against drug-resistant FLT3 mutations (*Figure 6A*). Since both staralogs are Type I inhibitors, they are not expected to suffer from the Type II-resistance model at the GK (F691L) and especially activation loop (D835) mutations observed in the clinic (*Smith et al., 2012*). The reduced potency of Star 27 against FLT3 relative to PKC412 in Ba/F3 cells (*Figure 6A*) is mitigated by the more relevant comparison in human cells (*Figure 2A*) as well as the predicted increase in maximum tolerated dosing for clinical candidates avoiding the KIT anti-target. Robustness against emergent mutations is a desirable feature of PKC412 but its pan-Class III RTK inhibition (especially KIT) is likely a major limitation addressed here. Remarkably, the addition of a single methyl group ($CH_3$) likely caused the observed decrease in toxicity.

The strategy reported here for gaining selectivity for FLT3 inhibition over KIT and other Class III RTKs may have importance in other hematopoietic pathology as well. FLT3 has been shown to be important for classical DC maintenance and necessary for inflammatory DCs (*Ramos et al., 2014*). Consequently, FLT3 has been proposed to be a drug target for autoimmunity and inflammation, particularly atherosclerosis. The staralog, CEP701, has been investigated in this context but it is expected that its promiscuity, particularly among KIT and CSF1R, will hamper clinical development (*Dai et al., 2002*). Star 27 should enable a reduction in FLT3-associated inflammation and autoimmunity while allowing KIT and CSF1R to compensate with classical DC maintenance. Furthermore, the generality of our electrostatic Phe GK inhibition coupled with Thr GK avoidance model (*Figures 1D, 5*) may guide future medicinal chemistry efforts, while our in vivo study shows any promiscuity seen in *Figure 1E* does not compromise Star 27's efficaciousness in a well-validated in vivo model of hematopoiesis (*Davidson and Zon, 2004*; *de Jong and Zon, 2005*; *He et al., 2014*).

Recent clinical experience with targeted inhibition of mutated kinases in cancer suggests that efficacy is driven by maximal pathway inhibition. In the case of vemurafenib, <80% pathway inhibition in BRAF (V600E) mutant melanoma afforded no tumor shrinkage, while >90% inhibition showed profound clinical benefit (*Bollag et al., 2010*). The challenge is that inhibitors of the oncogene also exhibit off-target effects on closely related kinases that inevitably lead to dose limiting toxicities. The goal of advancing improved molecular-targeted therapies is to identify the key anti-targets that drive toxicity and develop agents that avoid these effects. Our work highlights the synthetic lethal toxicity of an anti-target (KIT) when the oncogenic kinase (FLT3) is inhibited. As more complete understanding of kinase networks emerge it may be possible to delineate the anti-targets to avoid in numerous disease settings to allow improved kinase inhibitor design that can lead to greater ability to inhibit the disease causing pathway activation while avoiding systemic toxicities.

## Materials and methods

### Inhibitors
PKC412 was purchased from Selleckchem (Houston, TX).

### Chemical synthesis
Staralogs (*Figure 1B*) were synthesized according to precedent (*Lopez et al., 2013*). This protocol was adapted for synthesis of Star 27. Materials obtained commercially were reagent grade and were used without further purification. 1H NMR and 13C NMR spectra were recorded on Varian 400 (Palo Alto, CA) or Brucker 500 (Billerica, MA) spectrometers at 400 and 125 MHz, respectively. Low resolution mass spectra (LC/ESI-MS) were recorded on a Waters Micromass ZQ equipped with a Waters 2695 Separations Module and a XTerra MS C18 3.5 mm column (Waters). Reactions were monitored by thin layer chromatography (TLC), using Merck silica gel 60 F254 glass plates (0.25 mm thick). Flash chromatography was conducted with Merck silica gel 60 (230–400 mesh).

### In vitro kinase assays
For *Figure 1B* purified FLT3 WT was diluted in kinase reaction buffer (10 mM HEPES [pH 7.6], 10 mM $MgCl_2$, 0.2 mM DTT, 1 mg/ml BSA) to a concentration of 2 nM and pre-incubated with 2.5% (vol/vol) DMSO, 100 µM peptide (sequence EAIYAAPFKKK [Abltide]), and varying concentrations of inhibitor from 20 µM by fourths down to 1.2 nM for 10 min pre-incubation. Kinase reactions were initiated by the addition of 100 µM cold ATP supplemented with 2.5 µCi γ32P ATP per well and allowed to proceed at RT. At 15 min, 3 µL of the reactions were spotted onto phosphocellulose sheets (P81,

Whatman) and subsequently soaked in wash buffer (1.0% [vol/vol] phosphoric acid) at least five times for 5 min each. The sheets were then dried, and transferred radioactivity was measured by phosphorimaging using a Typhoon scanner (Molecular Dynamics). Radioactive counts were quantified using ImageQuant software (GE Healthcare Biosciences, Pittsburgh, PA), and titration data were fit to a sigmoidal dose response to derive $IC_{50}$ values using the Prism 4.0 software package. Experiments were performed 2–4 times, each in triplicate, with eight dosages, to derive standard error of the mean values. For *Figure 1C*, values obtained at Reaction Biology Corporation (Malvern, PA). Briefly, specific kinase/substrate pairs were prepared in reaction buffer; 20 mM Hepes pH 7.5, 10 mM $MgCl_2$, 1 mM EGTA, 0.02% Brij35, 0.02 mg/ml BSA, 0.1 mM $Na_3VO_4$, 2 mM DTT, 1% DMSO. Compounds were delivered into the reaction, followed 20 min later by addition of a mixture of ATP (Sigma, St. Louis, MO) and 33P ATP (PerkinElmer) to a final concentration approximating each kinase's $K_{m\text{-}ATP}$ (FLT3: 50 µM; KIT: 150 µM; CSF1R: 150 µM; PDGFRa: 5 µM; PDGFRb: 50 µM). Reactions were carried out at 25°C for 120 min, followed by spotting of the reactions onto P81 ion exchange filter paper (Whatman). Unbound phosphate was removed by extensive washing of filters in 0.75% phosphoric acid. After subtraction of background derived from control reactions containing inactive enzyme, kinase activity data were expressed as the percent remaining kinase activity in test samples compared to vehicle (dimethyl sulfoxide) reactions. $IC_{50}$ values and curve fits were obtained using Prism (GraphPad Software). Data obtained in singlicate with two biological replicates and $IC_{50}$ values presented ±SEM in *Figure 1C*.

## Kinome profiling

319 kinases were tested in a duplicate single dose (1 µM) format using a $^0$P-labeled ATP activity assay performed by Reaction Biology Corp. Assays performed at identical conditions to those of PKC412 (*Anastassiadis et al., 2011*).

## Cell lines

Stable Ba/F3 lines were generated by retroviral spinfection with the appropriate mutated plasmid as previously described (*Smith et al., 2012*).

## Cell-viability assay

Exponentially growing cells ($5 \times 10^3$ cells per well) were plated in each well of a 96-well plate with 0.1 ml of RPMI 1640 + 10% (vol/vol) FCS containing the appropriate concentration of drug in triplicate, and cell viability was assessed after 48 hr as previously described (*Smith et al., 2012*). $Log_{10}$-fold selectivity heat maps for *Figures 1C, 6A*: briefly, IC50 values were calculated as shown in figures, then ratios of RTK/FLT3 (*Figure 1C*) or point mutant/BaF3 FLT3 ITD (*Figure 6A*) were calculated. $Log_{10}$-scale transformation of ratios, followed by conditional formatting using xcel spreadsheet software yielded the colored diagrams shown.

## Assessment of caspase-3 activation

Exponentially growing cells were plated in the presence of PKC412 or Star 27 in RPMI + 10% (vol/vol) FCS for 48 hr. Cells were fixed with 4% (vol/vol) paraformaldehyde (Electron Microscopy Sciences) and permeabilized with 100% (vol/vol) methanol (Electron Microscopy Sciences, Hatfield, PA) followed by staining with a FITC-conjugated antiactive caspase-3 antibody (BD Pharmingen, San Jose, CA). Cells were run on a BD LSRFortessa cell analyzer, and data were analyzed using FlowJo (Tree Star Inc., Ashland, OR). Percentage of live cells was determined by negative staining for activated caspase-3 (see *Figure 2B,C*).

## Immunoblotting

Exponentially growing Molm14, HB119, or Ba/F3 cells stably expressing mutant isoforms were plated in RPMI medium 1640 + 10% (vol/vol) FCS supplemented with PKC412 or Star 27 at the indicated concentration. HMC1.2 cells were cultured and treated in IMDM + 10% FCS. After a 90-min incubation, the cells were washed in PBS, lysed, and processed as previously described. Immunoblotting was performed using anti-phospho-FLT3, anti-phospho-KIT, anti-phospho-STAT5, anti-STAT5, anti-phospho-ERK, anti-ERK, anti-phospho-S6, anti-S6, anti-KIT (Cell Signaling, Beverley, MA) and anti-FLT3 S18 antibody (Santa Cruz Biotechnology, Santa Cruz, CA).

## Zebrafish maintenance and embryo collection

Wild-type (WT) zebrafish were maintained under standard conditions, and embryos were staged as described (*He et al., 2014*). The study was approved by the Committee of the Use of Live Animals for Teaching and Research in The University of Hong Kong.

## Assessment of FLT3 colony assays in primary patient blasts

Primary AML blood samples and/or marrow aspirates were obtained on an IRB-approved protocol at the University of California, San Francisco. Informed consent was obtained in accordance with the Declaration of Helsinki. Mononuclear cells were purified by density centrifugation (Ficoll–Paque Plus, GE Healthsciences) before cryopreservation in 10% (vol/vol) DMSO or FCS (primary ITD and D835-mutant assays, *Figure 3A–C*) or immediate use in assays (Normal BM and SPB, *Figure 3D–F*). Normal BM and stimulated peripheral blood (SPB) samples were collected from donors at the UCSF oncology/hematology division. Cells were then suspended in MethoCult methylcellulose (Stem Cell Technologies, product no. H4435 Enriched, Canada) in 15-ml Falcon tubes, vigorously vortexed, bubbles settled over 10 min. Triplicates then plated via blunt-nosed syringe (1.1 ml each) into 3-cm dishes, avoiding bubble generation, and each dish placed in a larger 10-cm dish with water trough in an incubator (37°C, 5% $CO_2$) for 13–14 days. Colonies were counted individually using traditional microscopy. CFU and BFU types reported separately (CFU-GEMM colonies counted as one of each). Results shown for FLT3 ITD primary blasts (*Figure 3B*) are a combination of leukemic blast colonies and BFU colonies.

## Generation of mutants

Mutations isolated in the screen were engineered into pMSCVpuroFLT3–ITD by QuikChange mutagenesis (Stratagene, La Jolla, CA) as previously described (*Smith et al., 2012*).

## Acknowledgements

We gratefully acknowledge the Howard Hughes Medical Institute (KMS), the Leukemia and Lymphoma Society (CCS, EAL, KL, NPS), the National Institute of Health (NIHGMS R012R01EB001987, 1R01GM107671-01) (KMS), and the Waxman Foundation (KMS). The work on zebrafish (*Figure 4*) was supported by the General Research Fund (HKU771110 and HKU729809M) and the Zebrafish Core Facility at the Li Ka Shing Faculty of Medicine, and the Li Shu Fan Medical Foundation Professorship in Haematology (AYHL). We thank Dr Masanori Okaniwa for assistance with modeling studies (*Figure 5*), and Evan Massi for *Figure 6B*.

## Additional information

### Funding

| Funder | Grant reference number | Author |
|---|---|---|
| Howard Hughes Medical Institute | | Kevan M Shokat |
| Leukemia and Lymphoma Society | LLS TRP 6081-12 | Neil P Shah |
| Samuel Waxman Cancer Research Foundation | | Kevan M Shokat |
| National Institute of General Medical Sciences | NIHGMS R012R01EB001987 | Kevan M Shokat |
| National Cancer Institute | NCI R01 CA166616-01 | Neil P Shah |

The funders had no role in study design, data collection and interpretation, or the decision to submit the work for publication.

### Author contributions

AAW, Conception and design, Acquisition of data, Analysis and interpretation of data, Drafting or revising the article, Contributed unpublished essential data or reagents; MSL, Conception and design, Analysis and interpretation of data, Drafting or revising the article, Contributed unpublished essential data or reagents; EAL, B-LH, Acquisition of data, Analysis and interpretation of data; KL, Acquisition of data, Contributed unpublished essential data or reagents; AYHL, Analysis and interpretation of data, Contributed unpublished essential data or reagents; CCS, KMS, Conception and design, Analysis and interpretation of data, Drafting or revising the article; NPS, Drafting or revising the article

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
