## [Decision Letter]

Thank you for sending your work entitled ”Overcoming Myelosuppression Due to Synthetic Lethal Toxicity for FLT3-Targeted Acute Myeloid Leukemia Therapy” for consideration at *eLife*. Your article has been favorably evaluated by Charles Sawyers (Senior editor), a Reviewing editor, and 3 reviewers.

The Reviewing editor and the reviewers discussed their comments before we reached this decision, and the Reviewing editor has assembled the following comments to help you prepare a revised submission.

The reviewers all found your manuscript to be interesting and significant. Some criticisms were also noted, however. Both Reviewers 1 and 2 expressed an interest in seeing a broader selectivity analysis for STAR27, and Reviewer 3 raised some valid points about the cellular models used to mimic in vivo hematopoiesis and whether they accurately predict in vivo toxicity, or lack thereof (see specific comments below). We would be interested in receiving a suitably revised manuscript that addresses these comments.

*Reviewer 1 major comment*:

Overall, I think the rationale for the development of a FLT3-selective inhibitor (STAR27) is sound and the authors have been successful in achieving this goal. The level of biological characterization of STAR27 is appropriate and this reagent will likely prove useful for comparing the consequences of FLT3 versus FLT3/KIT inhibition. I believe that the impact of this work is sufficient enough to warrant publication in *eLife*. My only suggestion is that the authors discuss what the likely kinome-wide profile of STAR27 is. The Shokat Lab has shown that bisindolyl lactams with large substituents can be very selective for Ala or Gly gatekeeper-containing kinases. In this case, they are complementing the Phe residue of FLT3 with a small methyl residue in order to obtain selectivity for FLT3 over KIT (and other kinases that contain Thr or Val residues). This could potentially greatly increase the number of off-targets that this inhibitor acts upon. As STAR27 does not seem to be toxic, any off-targets do not seem to be too detrimental but it would nice if the authors speculated on how promiscuous STAR27 is based on their experience with other STAR analogs.

*Reviewer 2 major comment*:

It would have been nice to compare PKC412 and the new compound in at least a few additional kinase assays covering members outside the presented class III RTKs. Did the authors also explore the (R)-enantiomer of cpd 27?

*Reviewer 3 major comments*:

I am not sure how much stock one can take into ex vivo colony assays as a surrogate for in vivo hematopoiesis. Ideally I would like to see in vivo therapy in a mouse or xeno model, or at worst ex vivo treatment of human cells f/b transplant into NSG mice comparing the different drugs. These sorts of studies would increase the strength of these findings.

What is the data that Kit inhibition is responsible for the hematopoietic toxicity? Dasatinib inhibits Kit at clinically relevant concentrations, as does imatinib, without severe cytopenias. Is the toxicity due to dual flt3/kit inhibition? If so can the authors show that adding a selective Kit inhibitor to their Flt3 inhibitor results in inhibition of colony formation?

---

## [Author Response]

Reviewer 1 major comment:

*Overall, I think the rationale for the development of a FLT3-selective inhibitor (STAR27) is sound and the authors have been successful in achieving this goal. The level of biological characterization of STAR27 is appropriate and this reagent will likely prove useful for comparing the consequences of FLT3 versus FLT3/KIT inhibition. I believe that the impact of this work is sufficient enough to warrant publication in eLife. My only suggestion is that the authors discuss what the likely kinome-wide profile of STAR27 is. The Shokat Lab has shown that bisindolyl lactams with large substituents can be very selective for Ala or Gly gatekeeper-containing kinases. In this case, they are complementing the Phe residue of FLT3 with a small methyl residue in order to obtain selectivity for FLT3 over KIT (and other kinases that contain Thr or Val residues). This could potentially greatly increase the number of off-targets that this inhibitor acts upon. As STAR27 does not seem to be toxic, any off-targets do not seem to be too detrimental but it would nice if the authors speculated on how promiscuous STAR27 is based on their experience with other STAR analogs*.

We performed a kinome-wide profiling of Star 27 (new Figure 1). The broad kinome selectivity reveals that Star 27 is particularly able to distinguish between the Class III RTK family, of which FLT3 and KIT are members. This broad selectivity screening supports our electrostatic model (see Figure 5), explaining Star 27’s selectivity for Phe gatekeepers (GK) and avoidance of Thr GKs. As Reviewer #1 noted, the methyl subsituent on Star 27 is not large enough to limit binding to the broader kinome.

Reviewer 2 major comment:

*It would have been nice to compare PKC412 and the new compound in at least a few additional kinase assays covering members outside the presented class III RTKs. Did the authors also explore the (R)-enantiomer of cpd 27*?

We did not test this in this study because our previous work showed that this enantiomer lost all activity (see [44]). Our modeling is consistent with this strong structure activity relationship.

Reviewer 3 major comments:

*I am not sure how much stock one can take into ex vivo colony assays as a surrogate for in vivo hematopoiesis. Ideally I would like to see in vivo therapy in a mouse or xeno model, or at worst ex vivo treatment of human cells f/b transplant into NSG mice comparing the different drugs. These sorts of studies would increase the strength of these findings*.

To assess toxicity and efficacy in vivo we initiated a collaboration with Prof. Leung, from the University of Hong Kong, who has developed a recently validated zebrafish model for studying FLT3-ITD-driven AML, and recapitulates findings in the clinic (20). Zebrafish have been shown to be an excellent in vivo model of hematopoiesis (see Davidson et al, 2004; de Jong et al, 2005; [20]). We tested Star 27 head-to-head with PKC412 in WT animals, and assessed toxicity at both 3 days post fertilization (dpf, gross qualitative effect on heart and tail morphology; new Figure 4), and 30 hours post fertilization (hpf, staining as a quantitation of KIT-associated myelopoiesis; new Figure 4). These studies showed clear myelosuppression in PKC412 treated animals with no toxicity in Star 27 treatment groups compared to vehicle treated animals. To assess therapeutic efficacy we treated embryos injected with FLT3-ITD with both drugs. Both drugs greatly converted both “intermediate” and “severe” binned neutrophilic blast groups to “normal” embryos at 30 hpf, consistent with their equipotency in our other models of AML (new Figure 4). Prof Leung and Dr He have been added as authors.

*What is the data that Kit inhibition is responsible for the hematopoietic toxicity*?

KIT inhibition alone is not toxic, but in the context of KIT + FLT3 inhibition, both genetic evidence (double knockout—Lemischka, [29]) and our pharmacological data with Star 27 reveal the synthetic toxicity of inhibition of these two kinases.

*Dasatinib inhibits Kit at clinically relevant concentrations, as does imatinib, without severe cytopenias. Is the toxicity due to dual flt3/kit inhibition? If so can the authors show that adding a selective Kit inhibitor to their Flt3 inhibitor results in inhibition of colony formation*?

We were able to harvest normal bone marrow cells from a donor and tried combining multiple doses of Star 27 with multiple doses of either imatinib or dasatinib in the same colony assay format as we used in original Figure 3. Unfortunately this particular donor cell harvest did not yield enough blasts to score the experiment with sufficient statistical power. We have not been able to secure another donor in time for resubmission. Therefore we will not be able to include this requested experiment.